# Serum IgG titers to periodontal pathogens predict 3-month outcome in ischemic stroke patients

Shiro Aoki [1], Naohisa Hosomi[2,3]*, Hiromi Nishi[4], Masahiro Nakamori[5], Tomohisa Nezu[1], Yuji Shiga[1], Naoto Kinoshita[1], Hiroki Ueno[1], Kenichi Ishikawa[1,5], Eiji Imamura[5], Tomoaki Shintani[6], Hiroki Ohge[7], Hiroyuki Kawaguchi[4], Hidemi Kurihara[6,8], Hirofumi Maruyama[1]

1 Department of Clinical Neuroscience and Therapeutics, Graduate School of Biomedical and Health Sciences, Hiroshima University, Hiroshima, Japan, 2 Department of Neurology, Chikamori Hospital, Kochi, Japan, 3 Department of Disease Model, Research Institute of Radiation Biology and Medicine, Hiroshima University, Hiroshima, Japan, 4 Department of General Dentistry, Hiroshima University Hospital, Hiroshima, Japan, 5 Department of Neurology, Suiseikai Kajikawa Hospital, Hiroshima, Japan, 6 Center of Oral Examination, Hiroshima University Hospital, Hiroshima, Japan, 7 Department of Infectious Diseases, Hiroshima University Hospital, Hiroshima, Japan, 8 Department of Periodontal Medicine, Division of Applied Life Sciences, Institute of Biomedical and Health Sciences, Hiroshima University, Hiroshima, Japan

* nhosomi@hiroshima-u.ac.jp

**Data Availability Statement:** All relevant data are within the paper and its Supporting Information files.

## Abstract

Several cohort studies have shown that periodontal disease is associated with an increased risk for stroke. However, it remains unclear whether serum antibody titers for a specific periodontal pathogen are associated with outcome after ischemic stroke, and which kinds of pathogens are associated with ischemic stroke. We examined the relationship between serum IgG titers to periodontal pathogens and outcome in ischemic stroke patients. A total of 445 patients with acute ischemic stroke (194 female [44.0%], mean age 71.9±12.3 years) were registered in this study. Serum IgG titers to 9 periodontal pathogens (*Porphyromonas gingivalis*, *Aggregatibacter actinomycetemcomitans*, *Prevotella intermedia*, *Prevotella nigrescens*, *Fusobacterium nucleatum*, *Treponema denticola*, *Tannerella forsythensis*, *Campylobacter rectus*, *Eikenella corrodens*) were evaluated using the enzyme-linked immunosorbent assay (ELISA) method. An unfavorable outcome was defined as a 3 or higher on the modified Rankin Scale. The proportion of patients with unfavorable outcome was 25.4% (113 patients). Based on multivariate logistic regression analysis, numbers of IgG antibodies positive for periodontal pathogens (odds ratio 1.20, 95% CI 1.02–1.41, p = 0.03) were independent predictors of unfavorable outcome in ischemic stroke patients.

## Introduction

Periodontal disease is a persistent bacterial infection causing chronic inflammation in periodontal tissues that is characterized by loss of connective tissue and alveolar bone support, leading to tooth loss [1]. The prevalence of periodontal disease is very high. Approximately >90% of the world population have mild to advanced periodontal disease.

**Funding:** This study was supported by research grants from JSPS KAKENHI Grant Number (17K17350, 17K17907, and 18K10746). The funders had no role in study design, data collection and analysis, decision to publish, or preparation of the manuscript. There was no additional external funding received for this study.

**Competing interests:** The authors have declared that no competing interests exist.

Several cohort studies have shown that periodontal disease is associated with an increased risk of stroke [2–5]. A recent meta-analysis of 2 cohort studies found that periodontal disease increased the risk of ischemic stroke by 1.6-fold [6]. Furthermore, the Atherosclerosis Risk in Communities (ARIC) cohort study of 10362 stroke-free participants over a 15-year follow-up period demonstrated that periodontal disease was significantly associated with incidence of cardioembolic and thrombotic stroke, and regular dental care was related to lower adjusted stroke risk [7]. Periodontal disease has been established as one of the risk factors of ischemic stroke.

Measurement of serum antibody titers to a specific periodontal pathogen is considered to reflect its involvement in the disease process and has been used as one of the criteria required to identify causative organisms [8]. Recently, serum antibody titers related to a specific periodontal pathogen have been shown to indicate the risk factor of systemic diseases, such as coronary heart disease, non-alcoholic fatty liver disease and Alzheimer's disease [9–11]. We also previously reported that serum antibody titers related to a specific periodontal pathogen were associated with atrial fibrillation and carotid artery atherosclerosis [12].

However, it remains unclear whether serum antibody titers to periodontal pathogens can predict outcome after ischemic stroke. In this study, we examined the relationship between serum IgG titers to periodontal pathogens and outcome in ischemic stroke patients.

## Materials and methods

### Subjects

Consecutive acute ischemic stroke patients who were admitted to Hiroshima University Hospital and Suiseikai Kajikawa Hospital from January 2013 to April 2016 were enrolled in this prospective study. The study protocols were approved by the ethics committee of Hiroshima University Hospital and Suiseikai Kajikawa Hospital, and the study was performed according to the guidelines of the national government based on the Helsinki Declaration of 1964. Written informed consent was obtained from all patients or their relatives. All data analyses were blinded.

### Data acquisition

Imaging analysis by computed tomography or magnetic resonance imaging was performed in all patients for diagnosis. Baseline clinical characteristic data, including age, sex, drinking and smoking habits, comorbidities (hypertension, diabetes mellitus, dyslipidemia, atrial fibrillation), past history of stroke, and C-reactive protein (CRP) levels were collected from all patients. Ischemic stroke subtypes were classified using the Trial of Org 10172 in Acute Stroke Treatment (TOAST) criteria [13] by stroke specialists. Stroke severity on admission was evaluated using the National Institutes of Health Stroke Scale (NIHSS) scores. Hypertension was defined as use of anti-hypertensive medication before admission or confirmed blood pressure of $\geq$140/90 mmHg at rest measured 2 weeks after onset. Diabetes mellitus was defined as glycated hemoglobin level of $\geq$6.5%, fasting blood glucose level of $\geq$126 mg/dL, or use of anti-diabetes medication. Dyslipidemia was defined as total cholesterol level of $\geq$220 mg/dL, low-density lipoprotein cholesterol level of $\geq$140 mg/dL, high-density lipoprotein cholesterol level of <40 mg/dL, triglyceride levels of $\geq$150 mg/dL, or use of anti-hyperlipidemia medication. Atrial fibrillation was defined as follows: (1) a history of sustained or paroxysmal atrial fibrillation or (2) atrial fibrillation detection on arrival or during admission. Renal function was calculated with the estimated glomerular filtration rate (eGFR) using a revised equation for the Japanese population as follows: eGFR (mL min−1 1.73 m−2) = 194 × (serum creatinine)−1.094 × (age)−0.287 × 0.739 (for women) [14]. Chronic kidney disease was defined as an eGFR<60

mL min−1 1.73 m−2. When we evaluated the 3-month outcome, patients who were disabled prior to stroke incidence (corresponding to premorbid modified Rankin scale [mRS] score ≥2) were excluded. An unfavorable 3-month outcome was defined as a 3 or higher on the mRS.

## Measurement of serum antibody titers to periodontal pathogens

Serum IgG antibody titers to periodontal pathogens were determined using enzyme-linked immunosorbent assay (ELISA) as described previously [10]. Serum samples were collected from the patients within 3 days after stroke occurrence, and were stored at −80˚C. Sonicated preparations of the following periodontal pathogens were used as bacterial antigens *Porphyromonas gingivalis*, *Aggregatibacter actinomycetemcomitans*, *Prevotella intermedia*, *Prevotella nigrescens*, *Fusobacterium nucleatum*, *Treponema denticola*, *Tannerella forsythia*, *Campylobacter rectus*, and *Eikenella corrodens*. These pathogens were representative periodontal pathogens that were previously reported to be associated with serum antibody titers and stroke outcome [15]. Serum from 5 healthy subjects were pooled and used for calibration. Using serial dilutions of pooled control serum, the standard reaction was defined based on ELISA units (EU), such that 100 EU corresponded to 1:3200 dilution of the calibrator sample. For statistical analysis, we used the common logarithms of serum IgG antibody titers. Cut-off points for reactivity (positive decision point) to each antigen were defined as more than 1 standard deviation (SD) from mean ELISA units (EU).

## Statistical analysis

Data are expressed as means ± SD or the median (minimum, maximum) for continuous variables and frequencies and percentages for discrete variables. Statistical analysis was performed using JMP 13 statistical software (SAS Institute Inc., Cary, NC, USA). The statistical significance of intergroup differences was assessed using unpaired t-test or Mann-Whitney's U-test (for continuous variables), or the Fisher exact test or $\chi^2$ test (for discrete variables) as appropriate. Baseline data in ischemic stroke patients were analyzed, and two-step strategies were employed to assess the relative importance of variables in their association with poor outcome, respectively, using least square linear regression analysis. First, a univariate analysis was performed. Then, a multi-factorial analysis was performed with selected factors that had $p<0.10$ on univariate analysis. We considered $p<0.05$ to be statistically significant. Receiver operating characteristic (ROC) curves were configured to establish cut-off points for numbers of positive for periodontal pathogens that optimally predicted the unfavorable outcome.

## Results

A total of 445 patients with acute ischemic stroke (194 female [44.0%], mean age 71.9±12.3 y) were registered in this study. Table 1 shows the baseline characteristics of all patients. The median NIHSS score was 3 (IQR 1–6), and the mean serum CRP was 0.57±1.60 mg/dL.

The proportion of patients with unfavorable outcome was 25.4% (113 patients). Table 2 shows the univariate analysis of baseline characteristics to determine associations with unfavorable outcome. Patients with unfavorable outcomes were significantly older and had a higher proportion of female sex, atrial fibrillation, and cardioembolism than those with favorable outcomes; patients with unfavorable outcomes also had a lower frequency of habitual drinking. The levels of serum CRP were significantly higher in patients with unfavorable outcomes than in those with favorable outcomes ($p<0.001$). The patients with unfavorable outcomes exhibited severe neurological deficits at admission.

**Table 1. Baseline characteristics of study subjects.**

|  | n = 445 |
|---|---|
| Age | 71.9±12.3 |
| Sex (female), n (%) | 194 (44.0) |
| Comorbidities |  |
| Hypertension, n (%) | 331 (74.4) |
| Diabetes mellitus, n (%) | 119 (26.7) |
| Dyslipidemia, n (%) | 201 (45.2) |
| Atrial fibrillation, n (%) | 88 (19.8) |
| Chronic kidney disease, n (%) | 148 (33.8) |
| Past history of stroke, n (%) | 127 (28.5) |
| Current smoker, n (%) | 184 (41.3) |
| Habitual drinker, n (%) | 209 (47.0) |
| NIHSS score, median (IQR) | 3 (1–6) |
| Serum CRP, mg/dL | 0.57±1.60 |
| Ischemic stroke subtypes |  |
| Cardioembloism, n (%) | 99 (22.2) |
| Large artery atherosclerosis, n (%) | 103 (23.1) |
| Small artery occlusion, n (%) | 91 (20.4) |
| Others, n (%) | 152 (34.2) |

NIHSS, National Institutes of Health Stroke scale; IQR, interquartile range; CRP, C-reactive protein.

Table 3 shows the univariate analysis of antibody titers to periodontal pathogens (positive for periodontal pathogens) to determine associations with unfavorable outcome. The patients with unfavorable outcomes had a significantly higher proportion of positive for *F. nucleatum* and *T. denticola* than those patients with favorable outcomes. Variables with p<0.10 on univariate analysis were forced to enter into the multivariate logistic regression analysis procedure. From these results, age, NIHSS scores on admission, and serum CRP were independently associated with unfavorable outcome. Furthermore, we confirmed that IgG antibody titers to *F. nucleatum* (odds ratio 7.64, 95% CI 3.54–16.91, p<0.001) were independent predictors of unfavorable outcome in ischemic stroke patients (S1 Table). There were no significant correlations between serum CRP and serum IgG titers to each periodontal pathogen.

Patients with unfavorable outcomes had greater numbers of positive for periodontal pathogens than those with favorable outcomes (p = 0.001). The optimal cut-off point of the numbers of positive for periodontal pathogens for predicting unfavorable outcome was ≥3, with a sensitivity of 27.4%, specificity of 89.3%, and area under the ROC curve of 0.695. The mRS at 3-months was significantly higher in patients with ≥3 positive for periodontal pathogens than in those with <3 positive for periodontal pathogens (p<0.001, Fig 1). Variables with p<0.10 on univariate analysis were forced to enter into the multivariate logistic regression analysis procedure in Table 4. The numbers of IgG antibodies positive for periodontal pathogens (odds ratio 1.20, 95% CI 1.02–1.41, p = 0.03) were independent predictors of unfavorable outcome in ischemic stroke patients.

## Discussion

We demonstrated that serum IgG titers to periodontal pathogens predict 3-month outcome in ischemic stroke patients. In particular, IgG antibody titers to *F. nucleatum* and numbers of

**Table 2. Univariate analysis of baseline characteristics to determine associations with unfavorable outcome.**

| Factors | Favorable (n = 332) | Unfavorable (n = 113) | p-value |
|---|---|---|---|
| Age | 70.0±12.2 | 77.2±11.1 | <0.001 |
| Sex (female), n (%) | 135 (40.7) | 61 (54.0) | 0.012 |
| Hypertension, n (%) | 246 (74.1) | 86 (76.1) | 0.24 |
| Diabetes mellitus, n (%) | 89 (26.8) | 30 (26.6) | 0.96 |
| Dyslipidemia, n (%) | 156 (47.0) | 45 (39.8) | 0.18 |
| Atrial fibrillation, n (%) | 44 (13.3) | 44 (38.9) | <0.001 |
| Chronic kidney disease, n (%) | 109 (32.8) | 39 (34.5) | 0.74 |
| Past history of stroke, n (%) | 98 (29.5) | 29 (25.7) | 0.43 |
| Current smoker, n (%) | 145 (43.7) | 39 (34.5) | 0.094 |
| Habitual drinker, n (%) | 167 (50.3) | 42 (37.2) | 0.023 |
| NIHSS score, median (IQR) | 2 (1–4) | 10 (5–20) | <0.001 |
| Serum CRP, mg/dL | 0.38±1.26 | 1.12±2.24 | <0.001 |
| Ischemic stroke subtypes | | | <0.001 |
| Cardioembloism, n (%) | 48 (14.5) | 51 (45.1) | |
| Large artery atherosclerosis, n (%) | 82 (24.7) | 21 (18.6) | |
| Small artery occlusion, n (%) | 86 (25.9) | 5 (4.4) | |
| Others, n (%) | 116 (34.9) | 36 (31.9) | |

NIHSS, National Institutes of Health Stroke scale; IQR, interquartile range; CRP, C-reactive protein.

IgG antibodies positive for periodontal pathogens were independent predictors of unfavorable outcome in ischemic stroke patients.

The associations between periodontal disease and ischemic stroke have been reported in several studies. The first National Health and Nutrition Examination Survey showed that periodontal disease is one of the risk factors of ischemic stroke [16]. The ARIC study confirmed an independent association between periodontal disease and incident ischemic stroke risk, particularly cardioembolic and thrombotic stroke subtypes [7]. Furthermore, some cohort studies have reported that dental care or periodontal disease treatment could reduce the incidence of ischemic stroke [7, 17, 18]. On the other hand, the influence of periodontal disease on outcome of ischemic stroke patients has not been established. We revealed a significant association

**Table 3. Univariate analysis of antibody titers to periodontal pathogens (positive for periodontal pathogens) to determine associations with unfavorable outcome.**

| Factors | Favorable (n = 332) | Unfavorable (n = 113) | p-value |
|---|---|---|---|
| Periodontal pathogens | | | |
| *Porphyromonas gingivalis*, n (%) | 39 (11.8) | 15 (13.3) | 0.67 |
| *Aggregatibacter actinomycetemcomitans*, n (%) | 35 (10.5) | 16 (14.2) | 0.30 |
| *Prevotella intermedia*, n (%) | 29 (8.7) | 17 (15.0) | 0.071 |
| *Prevotella nigrescens*, n (%) | 33 (9.9) | 11 (9.7) | 0.95 |
| *Fusobacterium nucleatum*, n (%) | 20 (6.0) | 31 (27.4) | <0.001 |
| *Treponema denticola*, n (%) | 43 (13.0) | 25 (22.1) | 0.023 |
| *Tannerella forsythia*, n (%) | 42 (12.7) | 18 (15.9) | 0.39 |
| *Campylobacter rectus*, n (%) | 29 (8.7) | 16 (14.2) | 0.093 |
| *Eikenella corrodens*, n (%) | 36 (10.8) | 16 (14.2) | 0.35 |
| Numbers of IgG antibodies positive for periodontal pathogens, median (IQR) | 0 (0–1) | 1 (0–3) | 0.001 |

IQR, interquartile range.

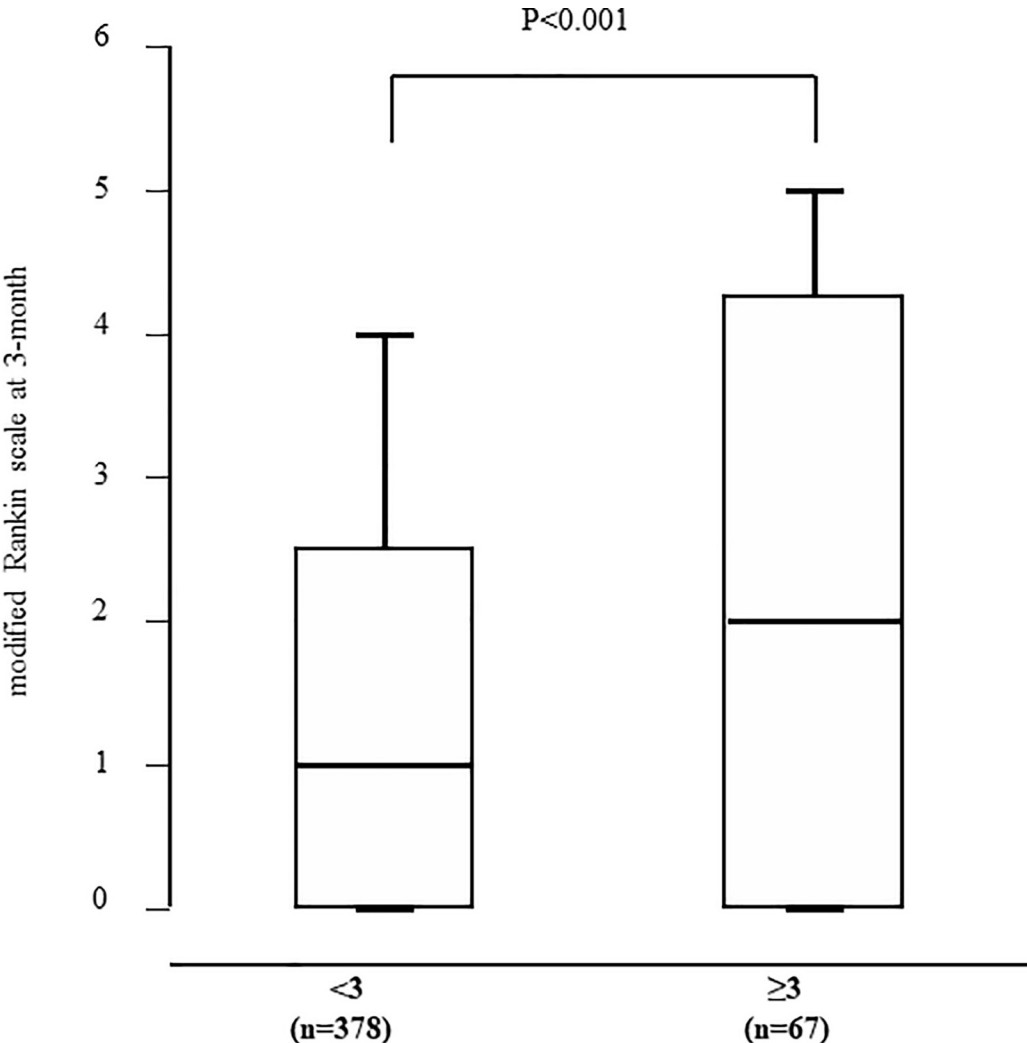

**Fig 1. Relationship between the mRS at 3-months and numbers of IgG antibodies positive for periodontal pathogens.** The mRS at 3-months was significantly higher in patients with ≥3 positive for periodontal pathogens than in those with <3 positive for periodontal pathogens.

between serum IgG titers to periodontal pathogens and outcome of ischemic stroke patients. There is a possibility that regular dental care not only reduces the incidence of ischemic stroke, but also prevents severe neurological deficits in the acute phase of ischemic stroke.

We used serum IgG titers to periodontal pathogens to investigate the association between periodontal disease and outcome of ischemic stroke. Circulating IgG levels against periodontal pathogens may be more accurate measures of periodontal infection and its severity in previous studies [19, 20]. There are several reports demonstrating the association between antibodies to periodontal pathogens and coronary heart disease [21–23]. However, there have been few studies investigating the association between antibodies to periodontal pathogens and ischemic stroke. Pussinen *et al.* showed that antibodies to *A. actinomycetemcomitans* and *P. gingivalis* are associated with incident stroke [24]. Hosomi *et al.* reported that the levels of serum anti-

**Table 4. Multivariate analyses to determine associations with unfavorable outcome.**

| Factors | Odds ratio | 95% CI | p-value |
|---|---|---|---|
| Age | 1.04 | 1.02–1.07 | 0.001 |
| Sex (female) | 1.31 | 0.67–2.58 | 0.44 |
| Current smoker | 1.21 | 0.62–2.37 | 0.58 |
| Habitual drinking | 0.43 | 0.22–0.81 | 0.008 |
| NIHSS score on admission | 1.22 | 1.15–1.29 | <0.001 |
| Cardioembolism | 1.33 | 0.66–2.59 | 0.41 |
| Serum CRP | 1.19 | 1.03–1.40 | 0.023 |
| Numbers of IgG antibodies positive for periodontal pathogens | 1.20 | 1.02–1.41 | 0.031 |

CI, confidence interval; NIHSS, National Institutes of Health Stroke scale; IQR, interquartile range, CRP: C-reactive protein.

*P. intermedia* antibody may be associated with atherothrombotic stroke [12]. We previously demonstrated that serum antibody titers to *F. nucleatum* were independent predictors of unfavorable outcome in all subtypes of stroke patients [15]. *F. nucleatum*, a Gram-negative anaerobe, is one of the most abundant species in the oral cavity, in both diseased and healthy individuals [25, 26]. It is considered to be a periodontal pathogen because it is frequently isolated from periodontitis lesions, produces a high number of tissue irritants, and often aggregates with other periodontal pathogens, as a bridge between early and late colonizers [27, 28]. It is frequently detected in atherosclerotic plaques, and is also one of the most common periodontal pathogens detected in ruptured cerebral aneurysm [29]. *F. nucleatum* elicits a variety of host responses, and is a potent stimulator of the inflammatory cytokines, IL-6, IL-8, and TNF α [30, 31].

In the present study, numbers of IgG antibodies positive for periodontal pathogens are independent predictors of unfavorable outcome in only ischemic stroke patients, not in hemorrhagic stroke patients. There is a possibility that an interaction between multiple periodontal pathogens may have negative effects on acute infarction. However, whether periodontal disease can modify acute ischemic brain damage is not fully understood. O'Boyle *et al.* showed that despite elevated systemic inflammation in periodontitis, oral inflammatory disease does not impact acute stroke pathology in terms of severity, determined primarily by infarct volume [32]. On the other hand, periodontal pathogens themselves are highly invasive; *P. gingivalis* can compromise and cross the blood brain barrier into the brain [33], and *Treponema* spp. have been speculated to enter the brain directly via peripheral trigeminal nerves [34]. *F.nucleatum* is also reported to be able to pass through the blood-brain barrier and has been found to be causative of brain abscesses in some case studies [35, 36]. Multiple periodontal pathogens infections are considered to cause higher-grade systemic inflammation. Several reports showed that high-grade systemic inflammation is deleterious in the context of ischemic stroke [37–39]. We speculate that this is a part of the reasons why numbers of IgG antibodies positive for periodontal pathogens are independent predictors of unfavorable outcome in ischemic stroke patients. Further studies are needed to clarify this point.

There are some limitations to our study. First, we did not evaluate the oral conditions such as tooth loss and grade of periodontal disease, socioeconomic status, and access to dental care in each patient. We cannot eliminate the possibility that these factors can influence the outcome in ischemic stroke patients. However, several reports have shown that IgG levels against periodontal pathogens are more accurate measures of periodontal infection and its severity. Takeuchi et al. showed that higher anti-*P. gingivalis* IgG levels were found in the periodontitis

group compared with the healthy control group [40]. Pussinen et al. found that ELISA is suitable for measuring antibodies to periodontal pathogens in large epidemiological studies in order to evaluate the role of periodontitis as a risk factor for other diseases [20]. Kudo et al. revealed that IgG titers of periodontitis patients were significantly higher than those of healthy controls, particularly in those with sites of probing depth over 4 mm [41]. In this way, IgG titers to periodontal pathogens are established as a surrogate marker of periodontal health. Moreover, we assessed the tooth status of 85 patients. The tooth status was evaluated by a dentist within 7 days after admission. The median numbers of remaining teeth were 23 (IQR 14–26), and treated teeth were 9 (IQR 4–14). Patients with unfavorable outcomes had significantly lower numbers of remaining teeth than patients with favorable outcomes. (20 [IQR 10–25] vs. 24 [IQR 16–27], p = 0.03). We found a positive correlation between the numbers of teeth lost and the numbers of positive for periodontal pathogens. Therefore, we believe that this limitation does not have a substantial influence on our results. Second, this is a cross-sectional observational study. It is difficult to elucidate the biological mechanisms responsible for the association between periodontal pathogens and ischemic stroke.

In conclusion, we demonstrated that numbers of IgG antibodies positive for periodontal pathogens were independent predictors of unfavorable outcome in ischemic stroke patients. It is possible that regular dental care not only reduces the incidence of ischemic stroke, but also prevents severe neurological deficits in the acute phase of ischemic stroke.

## Supporting information

**S1 Dataset.**
(XLSX)

**S1 Table. Multivariate analyses to determine associations with unfavorable outcome.**
(DOCX)

## Author Contributions

**Conceptualization:** Shiro Aoki, Naohisa Hosomi, Hiromi Nishi, Hidemi Kurihara.

**Data curation:** Shiro Aoki, Naohisa Hosomi, Hiromi Nishi, Masahiro Nakamori, Tomohisa Nezu.

**Formal analysis:** Shiro Aoki, Naohisa Hosomi.

**Funding acquisition:** Shiro Aoki, Naohisa Hosomi.

**Investigation:** Shiro Aoki, Naohisa Hosomi, Hiromi Nishi, Masahiro Nakamori, Tomohisa Nezu, Yuji Shiga, Naoto Kinoshita, Hiroki Ueno, Kenichi Ishikawa, Eiji Imamura, Tomoaki Shintani, Hiroyuki Kawaguchi.

**Methodology:** Hiromi Nishi, Tomoaki Shintani, Hiroki Ohge, Hiroyuki Kawaguchi, Hidemi Kurihara.

**Project administration:** Naohisa Hosomi, Hirofumi Maruyama.

**Supervision:** Hidemi Kurihara.

**Writing – original draft:** Shiro Aoki.

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
