## [Decision Letter · Decision Letter 0]

1 Jun 2020

PONE-D-20-09034

Serum IgG titers to periodontal pathogens predict 3-month outcome in ischemic stroke patients

PLOS ONE

Dear Dr. Hosomi,

Thank you for submitting your manuscript to PLOS ONE. After careful consideration, we feel that it has merit but does not fully meet PLOS ONE’s publication criteria as it currently stands. Therefore, we invite you to submit a revised version of the manuscript that addresses the points raised during the review process.

We look forward to receiving your revised manuscript.

Kind regards,

Kazunori Toyoda, MD, PhD

Academic Editor

PLOS ONE

Journal Requirements:

'The funders had no role in study design, data collection and analysis, decision to publish, or preparation of the manuscript.'

Reviewers' comments:

Reviewer's Responses to Questions

**Comments to the Author**

1. Is the manuscript technically sound, and do the data support the conclusions?

Reviewer #1: Partly

Reviewer #2: No

2. Has the statistical analysis been performed appropriately and rigorously? 

Reviewer #1: Yes

Reviewer #2: No

3. Have the authors made all data underlying the findings in their manuscript fully available?

Reviewer #1: Yes

Reviewer #2: No

4. Is the manuscript presented in an intelligible fashion and written in standard English?

Reviewer #1: Yes

Reviewer #2: Yes

5. Review Comments to the Author

Reviewer #1: Aoki et al. investigated whether serum antibody titers for a specific periodontal pathogen are associated with outcome after ischemic stroke, and which kinds of pathogens are associated with ischemic stroke using serum samples of 445 patients with acute ischemic stroke. The authors found that IgG antibody titers to Fusobacterium nucleatum and numbers of positive for periodontal pathogens were independent predictors of unfavorable outcome in ischemic stroke patients. The findings are potentially interesting but there are some interpretational flaws that should be addressed.

Major comments

1. The same group published a paper showing that serum IgG antibody titer to Fusobacterium nucleatum is associated with unfavorable outcome after ischemic and hemorrhagic stroke (n=534). The main result seems to be similar although the current manuscript analyzed data only from ischemic stroke (n=445). The authors should clarify that this manuscript does not overlap substantially with the paper by Nishi et al. in Clin Exp Immunol (reference No. 13). Otherwise, the current manuscript may be subject to duplicate publication.

2. The authors suggests that Fusobacterium nucleatum elicits host responses and expanded the damage in the ischemic penumbra during the acute phase of ischemic stroke. This is an overstatement. The inflammatory response can elicit systemic responses not restricted to the ischemic penumbra as shown in an experimental study. The authors should cite the following paper (O’Boyle et al. Int J Stroke 2020) and discuss the mechanism in details. https://doi.org/10.1177/1747493019834191

3. As discussed by the authors, the biggest limitation of this study is lack of oral assessment. There still remains a possibility that poor stroke outcome is associated with general oral (periodontal) health but not associated with specific periodontal pathogens if the elevated titers reflect periodontal health. The authors have to incorporate their own or published data that can deny the possibility.

Minor comments

In the Abstract, the name of bacteria should be fully spelled out.

Reviewer #2: In this cross-sectional study the authors predict IgG titers to periodontal pathogens predict 3-month outcome in ischemic stroke patients. In particular, IgG antibody titers to F. nucleatum

and numbers of positive for periodontal pathogens were independent predictors of

unfavorable outcome in ischemic stroke patients. The manuscript would benefit from following additional analyses:

1) PD linked with socioeconomic status and access to dental care. Inclusion of the variables in model.

2) Adjustment for multiple comparisons.

3) Some evidence to the hypothesis that PD may worsen ischemic penumbra

4) Animal data on F nucleatum, why other pathogens were not predictors of unfavorable outcome

5) Basic information on tooth status would be useful as well

6. PLOS authors have the option to publish the peer review history of their article (what does this mean?). If published, this will include your full peer review and any attached files.

Reviewer #1: No

Reviewer #2: Yes: Souvik Sen MD MS MPH

---

## [Author Response · Author response to Decision Letter 0]

23 Jun 2020

Jun 24, 2020

Dear Kazunori Toyoda, MD, PhD

We are most grateful to you and the reviewers for the critical comments and useful suggestions that have helped us to improve our paper considerably. As indicated in the responses as follow, we have taken all these comments and suggestions into account in the revised version of our paper.

Response to Reviewer #1

We wish to express our appreciation to the Reviewer for his or her insightful comments, which have helped us to improve the manuscript significantly.

1. The same group published a paper showing that serum IgG antibody titer to Fusobacterium nucleatum is associated with unfavorable outcome after ischemic and hemorrhagic stroke (n=534). The main result seems to be similar although the current manuscript analyzed data only from ischemic stroke (n=445). The authors should clarify that this manuscript does not overlap substantially with the paper by Nishi et al. in Clin Exp Immunol (reference No. 13). Otherwise, the current manuscript may be subject to duplicate publication.

Because the pathophysiology is different between ischemic stroke and hemorrhagic stroke, we believe that each characteristic becomes much clear by analyzing separately. In our study, numbers of positive for periodontal pathogens are independent predictors of unfavorable outcome in only ischemic stroke patients, not in hemorrhagic stroke patients. There is a possibility that an interaction of multiple periodontal pathogens may have negative effects on only acute infarction. On the other hand, IgG titers to A. actinomycetemcomitans, that do not impact on ischemic stroke patients at all, are an independent factor for predicting cerebral hemorrhage growth in hemorrhage stroke patients. In this way, IgG titers to periodontal pathogens indicate difference impacts between ischemic stroke patients and hemorrhagic stroke patients. Therefore, we think that the analysis which focuses on ischemic stroke patients has a major significance for an elucidation of relationship between periodontitis and stroke. 

We have put this information in page 18-19, line 246-258.

2. The authors suggest that Fusobacterium nucleatum elicits host responses and expanded the damage in the ischemic penumbra during the acute phase of ischemic stroke. This is an overstatement. The inflammatory response can elicit systemic responses not restricted to the ischemic penumbra as shown in an experimental study. The authors should cite the following paper (O’Boyle et al. Int J Stroke 2020) and discuss the mechanism in details.

O’Boyle et al. showed that despite elevated systemic inflammation in periodontitis, oral inflammatory disease does not impact acute stroke pathology in terms of severity, determined primarily by infarct volume. However, the pathogen of periodontitis is only P. gingivalis in their study. In our study, IgG titers to P. gingivalis did not have an effect on outcomes in ischemic stroke patients. On the other hand, F.nucleatum is reported to be able to pass through the blood-brain barrier and has been found to be causative of brain abscesses in some case studies, while it also has abilities to adhere to and invade host vascular endothelial cells via FadA adhesion molecules, as FadA binds to vascular endothelial-cadherin on the cell surface, which triggers breakdown of endothelial cell-to-cell junctions. Following passage through the blood-brain-barrier, F.nucleatum organisms attack vascular endothelial cells in blood vessels in the brain, which can induce endothelial permeability via loosened cell junctions. We hypothesize that these mechanisms induce expanding the damage in the ischemic penumbra during the acute phase of ischemic stroke. This result suggests that the inflammatory response differs depending on the pathogen of periodontitis.

We have put this information in page 19-20, line 269-284.

3. As discussed by the authors, the biggest limitation of this study is lack of oral assessment. There still remains a possibility that poor stroke outcome is associated with general oral (periodontal) health but not associated with specific periodontal pathogens if the elevated titers reflect periodontal health. The authors have to incorporate their own or published data that can deny the possibility.

It is true that we did not evaluate the oral conditions such as tooth loss and grade of periodontal disease in most patients. We cannot eliminate the possibility that these factors can influence the outcome in ischemic stroke patients. However, several reports have shown that IgG levels against periodontal pathogens are more accurate measures of periodontal infection and its severity. Takeuchi Y et al. showed that higher anti-P. gingivalis IgG levels were found in the periodontitis group compared with the healthy control group. Pussinen PJ et al. found that ELISA is suitable for measuring antibodies to periodontal pathogens in large epidemiological studies in order to evaluate the role of periodontitis as a risk factor for other diseases. Kudo C et al. revealed that IgG titers of periodontitis patients were significantly higher than those of healthy controls, especially in those with sites of probing depth over 4 mm. In this way, IgG titers to periodontal pathogens are established as a surrogate marker of periodontal health. Moreover, we assessed the tooth status of 85 patients. The tooth status was evaluated by a dentist within 7 day after admission. The median numbers of remaining tooth were 23 (IQR 14-26), and treated tooth was 9 (IQR 4-14). Patients with unfavorable outcome were significantly lower numbers of remaining tooth than patients with favorable outcome. (20 [IQR 10-25] vs. 24 [IQR 16-27], p=0.03). We found that a positive correlation between the numbers of tooth loss and the numbers of positive for periodontal pathogens. Therefore, we believe that this limitation does not have a substantial influence on our results. 

We have put this information in page 21-22, line 293-307.

4. In the Abstract, the name of bacteria should be fully spelled out.

We fully spelled out the name of periodontal pathogens in the abstract section.

Response to Reviewer #2

We wish to express our appreciation to the Reviewer for his or her insightful comments, which have helped us to improve the manuscript significantly.

1. PD linked with socioeconomic status and access to dental care. Inclusion of the variables in model.

In this study, we did not evaluate the socioeconomic status and access to dental care in each patient. We cannot eliminate the possibility that these factors can influence the outcome in ischemic stroke patients. However, several reports have shown that IgG levels against periodontal pathogens are more accurate measures of periodontal infection and its severity. Takeuchi Y et al. showed that higher anti-P. gingivalis IgG levels were found in the periodontitis group compared with the healthy control group. Pussinen PJ et al. found that ELISA is suitable for measuring antibodies to periodontal pathogens in large epidemiological studies in order to evaluate the role of periodontitis as a risk factor for other diseases. Kudo C et al. revealed that IgG titers of periodontitis patients were significantly higher than those of healthy controls, especially in those with sites of probing depth over 4 mm. In this way, IgG titers to periodontal pathogens are established as a surrogate marker of periodontal health. Moreover, we assessed the tooth status of 85 patients. The tooth status was evaluated by a dentist within 7 day after admission. The median numbers of remaining tooth were 23 (IQR 14-26), and treated tooth was 9 (IQR 4-14). Patients with unfavorable outcome were significantly lower numbers of remaining tooth than patients with favorable outcome. (20 [IQR 10-25] vs. 24 [IQR 16-27], p=0.03). We found that a positive correlation between the numbers of tooth loss and the numbers of positive for periodontal pathogens. Therefore, we believe that this limitation does not have a substantial influence on our results.

We have put this information in page 21-22, line 293-307.

2. Adjustment for multiple comparisons.

We appreciate for your suggestion. However, to our understand, we made no multiple comparison in our article. And, it is not appropriate to make multiple comparisons in this study, we believe. If you, further, suggest to make multiple comparisons and its adjustment, please let us know what kind of analysis you want us to add.

3. Some evidence to the hypothesis that PD may worsen ischemic penumbra.

O’Boyle et al. showed that despite elevated systemic inflammation in periodontitis, oral inflammatory disease does not impact acute stroke pathology in terms of severity, determined primarily by infarct volume. However, the pathogen of periodontitis is only P. gingivalis in their study. In our study, IgG titers to P. gingivalis did not have an effect on outcomes in ischemic stroke patients. On the other hand, F.nucleatum is reported to be able to pass through the blood-brain barrier and has been found to be causative of brain abscesses in some case studies, while it also has abilities to adhere to and invade host vascular endothelial cells via FadA adhesion molecules, as FadA binds to vascular endothelial-cadherin on the cell surface, which triggers breakdown of endothelial cell-to-cell junctions. Following passage through the blood-brain-barrier, F.nucleatum organisms attack vascular endothelial cells in blood vessels in the brain, which can induce endothelial permeability via loosened cell junctions. We hypothesize that these mechanisms induce expanding the damage in the ischemic penumbra during the acute phase of ischemic stroke. This result suggests that the inflammatory response differs depending on the pathogen of periodontitis.

We have put this information in page 19-20, line 269-284.

4. Animal data on F nucleatum, why other pathogens were not predictors of unfavorable outcome.

F.nucleatum is reported to be able to pass through the blood-brain barrier and has been found to be causative of brain abscesses in some case studies, while it also has abilities to adhere to and invade host vascular endothelial cells via FadA adhesion molecules, as FadA binds to vascular endothelial-cadherin on the cell surface, which triggers breakdown of endothelial cell-to-cell junctions. Following passage through the blood-brain-barrier, F.nucleatum organisms attack vascular endothelial cells in blood vessels in the brain, which can induce endothelial permeability via loosened cell junctions. We hypothesize that these mechanisms induce expanding the damage in the ischemic penumbra during the acute phase of ischemic stroke. This result suggests that the inflammatory response differs depending on the pathogen of periodontitis.

We have put this information in page 19-20, line 274-284.

5. Basic information on tooth status would be useful as well.

In this study, we assessed the tooth status of 85 patients. The tooth status was evaluated by a dentist within 7 day after admission. The median numbers of remaining tooth were 23 (IQR 14-26), and treated tooth was 9 (IQR 4-14). Patients with unfavorable outcome were significantly lower numbers of remaining tooth than patients with favorable outcome. (20 [IQR 10-25] vs. 24 [IQR 16-27], p=0.03). We found that a positive correlation between the numbers of tooth loss and the numbers of positive for periodontal pathogens.

We have put this information in page 21-22, line 301-307.

---

## [Decision Letter · Decision Letter 1]

7 Jul 2020

PONE-D-20-09034R1

Serum IgG titers to periodontal pathogens predict 3-month outcome in ischemic stroke patients

PLOS ONE

Dear Dr. Hosomi,

Thank you for submitting your manuscript to PLOS ONE. After careful consideration, we feel that it has merit but does not fully meet PLOS ONE’s publication criteria as it currently stands. Therefore, we invite you to submit a revised version of the manuscript that addresses the points raised during the review process.

We look forward to receiving your revised manuscript.

Kind regards,

Kazunori Toyoda, MD, PhD

Academic Editor

PLOS ONE

Reviewers' comments:

Reviewer's Responses to Questions

**Comments to the Author**

1. If the authors have adequately addressed your comments raised in a previous round of review and you feel that this manuscript is now acceptable for publication, you may indicate that here to bypass the “Comments to the Author” section, enter your conflict of interest statement in the “Confidential to Editor” section, and submit your "Accept" recommendation.

Reviewer #1: (No Response)

2. Is the manuscript technically sound, and do the data support the conclusions?

Reviewer #1: Yes

3. Has the statistical analysis been performed appropriately and rigorously? 

Reviewer #1: Yes

4. Have the authors made all data underlying the findings in their manuscript fully available?

Reviewer #1: Yes

5. Is the manuscript presented in an intelligible fashion and written in standard English?

Reviewer #1: No

6. Review Comments to the Author

Reviewer #1: The authors have responded to the concerns partially.

1. The authors have added some comments on the difference between ischemic and hemorrhagic stroke in the discussion (page 18). Cite the reference 13 in the discussion, and discuss more concretely by showing the results obtained in the reference 13 and the current paper.

2. Although the authors elaborated on the mechanism by which F. nucleatum attack the blood vessels. They hypothesized the mechanism as stated in the Discussion (page 20) but it was really an overstatement as this reviewer suggested earlier. Why can the authors say the bacteria expand the damage in the penumbra. In particular, why can the authors pinpoint the site of damage in the penumbra? Without any experimental evidence, one cannot overstate the underlying mechanisms.

3. The added texts should be thoroughly checked by a native English speaker.

7. PLOS authors have the option to publish the peer review history of their article (what does this mean?). If published, this will include your full peer review and any attached files.

Reviewer #1: No

---

## [Author Response · Author response to Decision Letter 1]

20 Jul 2020

Jul 20, 2020

Dear Kazunori Toyoda, MD, PhD

We are most grateful to you and the reviewers for the critical comments and useful suggestions that have helped us to improve our paper considerably. As indicated in the responses as follow, we have taken all these comments and suggestions into account in the revised version of our paper.

Response to Reviewer #1

We wish to express our appreciation to the Reviewer for his or her insightful comments, which have helped us to improve the manuscript significantly.

1. The authors have added some comments on the difference between ischemic and hemorrhagic stroke in the discussion (page 18). Cite the reference 13 in the discussion, and discuss more concretely by showing the results obtained in the reference 13 and the current paper.

Nishi et al. previously reported that IgG titers to a specific periodontal pathogen (F. nucleatum) were associated with unfavorable outcome after all subtypes of stroke. On the other hand, we performed the comprehensive analysis of relationship between IgG titers to whole periodontal pathogens and acute ischemic stroke. To clarify this difference, we deleted the description about relationship between IgG titers to F. nucleatum and outcome in Abstract section and result section (table 4 [model 1]). Additionally, we revised the fourth paragraph of Discussion section as follow underlined (page 18-19, line 247-264)

In the present study, numbers of IgG antibodies positive for periodontal pathogens are independent predictors of unfavorable outcome in only ischemic stroke patients, not in hemorrhagic stroke patients. There is a possibility that an interaction between multiple periodontal pathogens may have negative effects on acute infarction. However, whether periodontal disease can modify acute ischemic brain damage is not fully understood. O’Boyle et al. showed that despite elevated systemic inflammation in periodontitis, oral inflammatory disease does not impact acute stroke pathology in terms of severity, determined primarily by infarct volume [32]. On the other hand, periodontal pathogens themselves are highly invasive; P. gingivalis can compromise and cross the blood brain barrier into the brain [33], and Treponema spp. have been speculated to enter the brain directly via peripheral trigeminal nerves [34]. F.nucleatum is also reported to be able to pass through the blood-brain barrier and has been found to be causative of brain abscesses in some case studies [35, 36]. Multiple periodontal pathogens infections are considered to cause higher-grade systemic inflammation. Several reports showed that high-grade systemic inflammation is deleterious in the context of ischemic stroke [37-39]. We speculate that this is a part of the reasons why numbers of IgG antibodies positive for periodontal pathogens are independent predictors of unfavorable outcome in ischemic stroke patients. Further studies are needed to clarify this point.

2. Although the authors elaborated on the mechanism by which F. nucleatum attack the blood vessels. They hypothesized the mechanism as stated in the Discussion (page 20) but it was really an overstatement as this reviewer suggested earlier. Why can the authors say the bacteria expand the damage in the penumbra. In particular, why can the authors pinpoint the site of damage in the penumbra? Without any experimental evidence, one cannot overstate the underlying mechanisms.

As the reviewer indicated, the speculation that F. nucleatum elicits host responses and expanded the damage in the ischemic penumbra during the acute phase of ischemic stroke is an overstatement. We revised the fourth paragraph of Discussion section as follow underlined (page 18-19, line 247-264)

In the present study, numbers of IgG antibodies positive for periodontal pathogens are independent predictors of unfavorable outcome in only ischemic stroke patients, not in hemorrhagic stroke patients. There is a possibility that an interaction between multiple periodontal pathogens may have negative effects on acute infarction. However, whether periodontal disease can modify acute ischemic brain damage is not fully understood. O’Boyle et al. showed that despite elevated systemic inflammation in periodontitis, oral inflammatory disease does not impact acute stroke pathology in terms of severity, determined primarily by infarct volume [32]. On the other hand, periodontal pathogens themselves are highly invasive; P. gingivalis can compromise and cross the blood brain barrier into the brain [33], and Treponema spp. have been speculated to enter the brain directly via peripheral trigeminal nerves [34]. F.nucleatum is also reported to be able to pass through the blood-brain barrier and has been found to be causative of brain abscesses in some case studies [35, 36]. Multiple periodontal pathogens infections are considered to cause higher-grade systemic inflammation. Several reports showed that high-grade systemic inflammation is deleterious in the context of ischemic stroke [37-39]. We speculate that this is a part of the reasons why numbers of IgG antibodies positive for periodontal pathogens are independent predictors of unfavorable outcome in ischemic stroke patients. Further studies are needed to clarify this point.

3. The added texts should be thoroughly checked by a native English speaker.

As the reviewer indicated, the added texts were checked by a native English speaker.

---

## [Editor Report · Decision Letter 2]

22 Jul 2020

Serum IgG titers to periodontal pathogens predict 3-month outcome in ischemic stroke patients

PONE-D-20-09034R2

Dear Dr. Hosomi,

We’re pleased to inform you that your manuscript has been judged scientifically suitable for publication and will be formally accepted for publication once it meets all outstanding technical requirements.

Kind regards,

Kazunori Toyoda, MD, PhD

Academic Editor

PLOS ONE
---

## [Editor Report · Acceptance letter]

27 Jul 2020

PONE-D-20-09034R2 

Serum IgG titers to periodontal pathogens predict 3-month outcome in ischemic stroke patients 

Dear Dr. Hosomi:

I'm pleased to inform you that your manuscript has been deemed suitable for publication in PLOS ONE. Congratulations! Your manuscript is now with our production department. 

Kind regards, 

on behalf of

Dr. Kazunori Toyoda 

Academic Editor

PLOS ONE